# Productivity vs. Evenness in the U.S. Financial Market: A Business Ecosystem Perspective

**DOI:** 10.3390/e25071029

**Published:** 2023-07-07

**Authors:** Hugo Fort

**Affiliations:** Institute of Physics, Faculty of Science, Universidad de la República, Montevideo 11400, Uruguay; hugofortquijano@gmail.com or hugo@fisica.edu.uy

**Keywords:** business ecosystem, population dynamics, Shannon evenness, co-evolution in markets

## Abstract

This paper starts by presenting an empirical finding in the U.S. stock market: Between 2001 and 2021, high productivity was achieved when the Shannon evenness—measuring the inverse of concentration—dropped. Conversely, when the Shannon evenness soared, productivity plunged. The same inverse relationship between evenness and productivity has been observed in several ecosystems. This suggests explaining this result by adopting the business ecosystem perspective, i.e., regarding the tangle of interactions between companies as an ecological network, in which companies play the role of species. A useful strategy to model such ecological communities is through ensembles of synthetic communities of pairwise interacting species, whose dynamics is described by the Lotka–Volterra generalized equations. Each community is specified by a random interaction matrix whose elements are drawn from a uniform distribution centered around 0. It is shown that the inverse relationship between productivity and evenness can be generated by varying the strength of the interaction between companies. When the strength increases, productivity increases and simultaneously the market evenness decreases. Conversely, when the strength decreases, productivity decreases and evenness increases. This strength can be interpreted as reflecting the looseness of monetary policy, thus providing a link between interest rates and market structure.

## 1. Introduction

Neoclassical economics, which assumes investors behave with rational expectations in order to maintain an efficient market, is frequently at odds to explain the dynamics of markets. Instead, the agents in markets are not perfectly rational, but rather they are boundedly rational satisfiers [1]. The idiosyncrasies in human behavior make financial markets depart from the assumption of informational efficiency leading for example to excess volatility, i.e., financial markets change more than rational measures of value would suggest [2].

An alternative viewpoint is to regard financial markets as ecosystems with a tangle of interactions between companies, investors, clients, etc. Indeed, according to [3,4], companies are engaged in “competition for differential advantage” which gives firms a position in the marketplace known as an “ecological niche” [3]. Companies survive and grow in the marketplace depending on the actions and reactions of agents permanently adjusting their behavior to match environmental opportunities. Such an ongoing process is similar to the one that operates in ecological systems competing for scarce resources [5,6,7]. That is, a process of *co-evolution*, shared by markets and ecosystems, in which interdependent species or companies evolve in an endless reciprocal cycle—such that changes in species A set the stage for the natural selection of changes in species B—and vice versa [8]. Co-evolution occurs in different forms, antagonistic, e.g., predators and their prey, mutually competitive, e.g., different species sharing the same trophic level, or cooperative co-evolution, e.g., flowering plants and their pollinators [9].

Moore [10] introduced biological ecology as a metaphor for strategic thinking about business co-evolution and radically new cooperative/competitive relationships. In a similar vein, Farmer and Lo [11] regard markets as co-evolving ecologies of different strategies pursued by companies. These strategies are analogous to a biological species, and the amount of funds deployed by traders following a given strategy is analogous to the population of that species [11]. As the market evolves, the market shares of the inefficient companies decrease while the companies with greatest fitness capture market share. Therefore, companies often play the same role of selection units that species play in ecosystems.

The general goal in this paper is to use the above analogy between markets and ecosystems to better understand the forces that structure markets and determine their productivity. This includes the market responses to external shocks (analogous to environmental perturbations), such as expansive economic policies (analogous to nutrient enrichment), and the susceptibility of companies of being displaced by newcomer companies (species turnover in the case of ecosystems). Furthermore, the above analogy offers an opportunity to harness the potential of applying various powerful techniques from theoretical ecology to the fields of economics and finance. The specific primary objective of this study is to elucidate the inverse relationship detected between productivity and evenness within a set of firms encompassing the largest companies in the U.S. stock market. To accomplish this, we employ a combination of empirical evidence and theoretical modeling from ecology. Unraveling this relationship holds significant importance as it profoundly impacts the functioning of both markets and ecosystems. In fact, this ecological perspective allows us to use two central attributes which emerge from the co-evolution process of species in an ecological community, namely its productivity and its species diversity [12,13] to get insight into market dynamics. Both properties can be defined in several different ways in ecology. Productivity has been characterized by variables that range from direct estimates of energy flow to the ecosystem to accumulated biomass or biomass density (per area or volume) [14,15]. A common metric is the rate of generation of biomass in an ecosystem, usually expressed in units of mass per unit area per unit of time, such as grams per square meter per day [16]. In the case of agricultural crops, productivity is also commonly measured by the *total weight* per unit area [17], which is known as *crop yield* [18]. Diversity, in turn, involves concepts ranging from simplest concept of *species richness*, namely the number of species, to *evenness*, i.e., the measure of how similar species are in their abundance in an environment [19]. Indeed, species diversity is often intended as a combination of richness and evenness [20].

This study draws on and integrates elements of ecological science and economics, which is the scientific research program of ecological economics (EE), understood as “the relationship between ecosystems and economic systems in the broadest sense” [21]. In addition, it is transdisciplinary and uses methods and complex systems analysis [22].

## 2. The Business Ecosystem Perspective: Financial Markets as Ecosystems

The business ecosystem perspective refers to a framework or approach that views businesses and organizations as part of a larger interconnected system or ecosystem [10]. In the business ecosystem perspective, the focus is not solely on individual firms operating in isolation, but rather on understanding how they interact and mutually influence each other within the broader context of the ecosystem [23,24,25]. It recognizes that the success and sustainability of any given organization are influenced by the health and dynamics of the entire ecosystem in which it operates. Key features of the business ecosystem perspective include interconnectedness, collaborative relationships, and ecosystem dynamics. Interconnectedness recognizes that firms within the ecosystem are interconnected and depend on each other for resources, capabilities, and market opportunities. Actions and changes in one part of the ecosystem can have ripple effects on other entities within the system. Collaborative relationships refer for example to partnerships and alliances among different companies within the ecosystem. Through ecosystem dynamics, we understand that the perspective acknowledges that ecosystems are dynamic and subject to various forces and disruptions, rather than entities at equilibrium. New entrants, technological advancements, market shifts, or changes in regulatory environments can shape the competitive landscape and the overall dynamics of the ecosystem [10,24].

Using the analogy between ecosystems and markets, companies can be regarded as species and the market value (In this paper ‘market value’ is taken as synonym of *market capitalization*, i.e., the number of a company’s shares outstanding multiplied by the current price of a single share [26].) of a company as the abundance or biomass of a species [7,11,23,24,25]. Therefore, as in ecology, we consider as proxy for productivity a relative metric, corresponding to total returns—i.e., the rate of variation of the total market value. In addition, as it is carried out in agricultural sciences, we also consider an absolute metric, given by the total market value (analogous to crop yield). Likewise, as a measure of diversity, the *Shannon evenness* [27]—aka *Shannon equitability index* is used. This metric is widely used in ecology, for example, to measure the variation of the diversity of a community with a fixed number of species [28]. Notice that, in the same way as species evenness is highest when all species in a community have the same abundance, the market evenness is highest when all firms have the same market share. Market evenness is the opposite of concentration, which happens for example when a few disproportionately large firms dominate the returns of value weighted stock market indices such as the S&P500. The use of the concept of evenness and other diversity measures in economics was reviewed, for example, in [29]. Additionally, a comparison of ecological and economic measures of biodiversity was reviewed in [30]. Box 1 summarizes the correspondences between financial markets and community ecology.

Box 1Correspondences between financial markets and community ecology.Financial market
Community ecologyDenoted by
company 
 ↔  species
*i*

market value of a company 
 ↔  species biomass
*v_i_*

total market value 
 ↔  total biomass (all species)
*V*

total market return
 ↔  rate of variation of thetotal biomass
*R*

market share of a company 
 ↔  frequency of such species
*x_i_*

evenness (inverse of concentration) 
 ↔  evenness (species diversity)
*E*


Most natural ecological communities exist in a state of nonequilibrium where competitive equilibrium is prevented by several factors such as, for example, fluctuations in the physical and biotic environment [31]. The same happens in stock markets, where stock prices often do not settle down for long time but are driven by factors affecting supply and demand such as the economic environment, economic policies, market news, etc. In nonequilibrium ecological communities, although the number of coexisting competitors remains relatively stable, the level of diversity—measured by the evenness—varies. Indeed, a long-standing debate in ecology is that of how species diversity relates to the productivity of ecosystems (see for instance [32] or [33] and references therein).

Classical community ecology, developed by Lotka [34] and Volterra [35], has been the major descriptor of species interactions in the ecological literature for almost a century. The Lotka–Volterra generalized theory (LVGT) [36,37] rests on the assumption that species interactions play a major role in structuring an ecological community. The Lotka–Volterra generalized equations can be written in finite time as [37,38]:(1)$$v_{i}(t+1)-v_{i}(t)=r_{i}v_{i}(t)\left(1+\sum_{j=1}^{S}\alpha_{ij}v_{j}(t)\right),\;\;i=1,2,\cdots ,S.$$
where *i* denotes the species number; *v_i_*(*t*) stands for its biomass at time *t* and *r_i_* is the intrinsic growth rate of the species (dimension of time^−1^). Thus, a central ingredient of LVGT is the *pairwise interaction matrix*, *α_ij_*. whose element *ij* quantifies the effect of species *j* on the growth of species *i*. The resulting variation in pairwise species interactions determines biodiversity in a community [39], and thus it is able to yield species abundance distributions and biodiversity as a function of species-specific interaction parameters. By analogy, one way to approach the relationship between productivity as a function of evenness in financial markets is through LVGT. The problem is that estimating the interaction matrix *α_ij_* between companies is far from trivial. We will come back to this problem in Section 4.

## 3. Empirical Analysis

### 3.1. Dataset

The used dataset is based on the Fortune 100 list, i.e., a list of the top 100 public and privately held companies by revenues in the United States published by Fortune magazine [40]. From these 100 U.S. companies we selected those 78 *public* firms such that reported annual revenue and market cap from 1 January 2000 (see Table 1). Thus, the resulting dataset consists of time series for daily closing market values for each company, *v_i_*(*t*) (*i* =1, 2, …, 78), with *t* measured in days spanning 5536 days, from 1 January 2000 to 31 December 2021 [41]. The market value is a good firm size proxy; indeed, over the 27-year period of 1989–2015, it demonstrated providing high value relevance in predicting future returns [42].

According to the Federal Reserve [43], there were three recessions in this period:From the first to the third quarter of 2001, corresponding to the dot-com crash [44];From the fourth quarter of 2007 to the second quarter of 2009, associated with the “Subprime Mortgage Crisis” or the “Mortgage crisis” [45];Across the first and second quarters of 2020.

Hence, this sample covers two business cycles.

### 3.2. Variables

In this study, as mentioned, the three main global or aggregated variables considered are:

1. The total market value, *V*(*t*), which depends on time *t* (measured in days), i.e.,
(2)$$V(t)\equiv \;\sum_{i=1}^{78}v_{i}(t).$$

2. The total market return, *R*(*t*), given by the annual variation of *V*, i.e.,
(3)$$R(t)\equiv \sum_{j=1}^{S}\left(v_{i}(t+1)-v_{i}(t)\right).$$

In fact, I mainly consider the two abovementioned quantities adjusted by the annual consumer price index (CPI), respectively, denoted as *V_a_*(*t*) and *R_a_*(*t*), except as otherwise stated.

3. The Shannon evenness or Shannon equitability, *E*(*t*), defined as:(4)$$E(t)\equiv \frac{-\sum_{i=1}^{78}x_{i}(t)\ln x_{i}(t)}{\ln78},$$
where *x_i_*(*t*) is the market share of company *i* at day *t*, i.e.,
(5)$$x_{i}(t)=\frac{v_{i}(t)}{V(t)}.$$

This index is basically a normalized Shannon entropy, independent of the sample size (*N* = 78 in our particular study).

In addition, using daily data raises the problem of high-frequency variation of daily prices compared to the monthly, quarterly, or annual frequency which are much more relevant for the business ecosystem picture. To avoid this problem, the high frequency daily fluctuations were smoothed out by using moving averages over 252 stock trading days per year.

It is worth mentioning that the total (unadjusted) market value *V* of this set of companies at the end of the period was USD 18.9 trillion [40,41], and they represented at least 60% of the total New York Stock Exchange (NYSE) market cap in the period 2000–2021 [46]. Hence, as expected, *V* is strongly correlated with the S&P 500 index, as shown in Figure 1. This simply confirms that *V* for the selected set of companies serves as an aggregate measure of production to determine the business cycle chronology (working with the entire set of 2800 NYSE listed firms would be a daunting task).

As an additional check that that *E*(*t*), given by Equation (4), correctly reflects the market evenness of the whole U.S. stock market the Shannon evenness was computed through Equation (3) but taking subsets of the whole set of 78 firms, i.e., the top 20 companies, the top 30, etc. (and replacing in Equation (3) 78 by *N* = 20, 30, etc.). Figure 2 shows that the corresponding succession of curves of *E_N_* (*t*) converges towards the evenness *E*(*t*) for the whole set; for *N* ≥ 50 the curves are qualitatively very similar, while the curve *E*_70_ (*t*) only shows small departures from *E*(*t*). This is because adding companies with very low shares does not change much *E* since *x_i_* ln *x_i_* → 0 when *x_i_* → 0.

It was checked that other different metrics used to quantify the evenness, like inverse Simpson and the Gini–Simpson indices, provide qualitatively identical curves to the Shannon evenness. (See Appendix A)

### 3.3. Productivity vs. Evenness in the U.S. Stock Market

The relationship between *R_a_*(*t*), the most widely used metric to measure market productivity, and *E*(*t*) provides clear evidence of the inverse relationship between productivity vs. evenness. In Figure 3, the 21-year period is divided into three portions according to the behavior of the evenness *E* (full thick green curve). That is:A period of soaring *E*(*t*), from January 2001 to December 2007, (almost exactly coinciding with the first business cycle);A period of relatively smooth oscillations of *E*(*t*) around a high value, from January 2008 to December 2017;A period in which *E* plunged, from January 2018 to December 2021.

**Figure 3 entropy-25-01029-f003:**
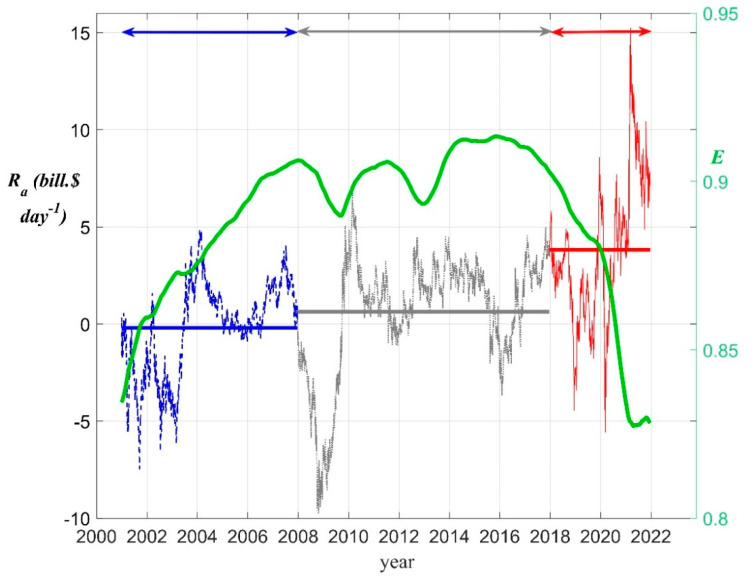
CPI adjusted total returns, *R_a_*, and Shannon evenness, *E*, along the period 2001–2022 [41]. Right axis: Shannon evenness *E* (full tick green curve). Left axis: *R_a_* plotted for three different periods depending on the behavior of *E*: 2001–2007 (dashed blue), characterized by soaring *E* and negative average *R_a_*; 2008–2017 (dotted gray), of roughly constant *E* and low average *R_a_*; January 2018 to December 2021, in which *E* plunged and the average *R_a_* was high. The horizontal segments correspond to the mean of *R_a_* along the respective period.

Notice that the *R_a_* averaged over these periods, indicated in Figure 3 by horizontal segments, was slightly negative (blue), slightly positive (gray), and high (red), respectively. In other words, during periods in which *E*(*t*) sharply decreased productivity was high, while in the other periods of soaring or high evenness, productivity was low.

The behavior of the absolute productivity metric, *V_a_*, is also enlightening. Figure 4 shows the trajectory of *V_a_* as a function of *E* from 2001 to 2021 together with some key financial events that occurred in this period. The three recessions divide the period in two business cycles, both characterized by *E* and *V_a_* moving in opposite directions:From the beginning of 2001 until the end of 2007 (portion of the trajectory in blue in Figure 4), in which E steadily increased, while *V_a_* ended in a slightly lower value.The 2009–2020 expansion (portion of the trajectory in red in Figure 4), which was the longest on record at 128 months—from July 2009 to February 2020—according to the Congressional Research Service (NBER 2022). This was a period in which, after some initial erratic movements, *V_a_* grew strongly and E considerable declined.

**Figure 4 entropy-25-01029-f004:**
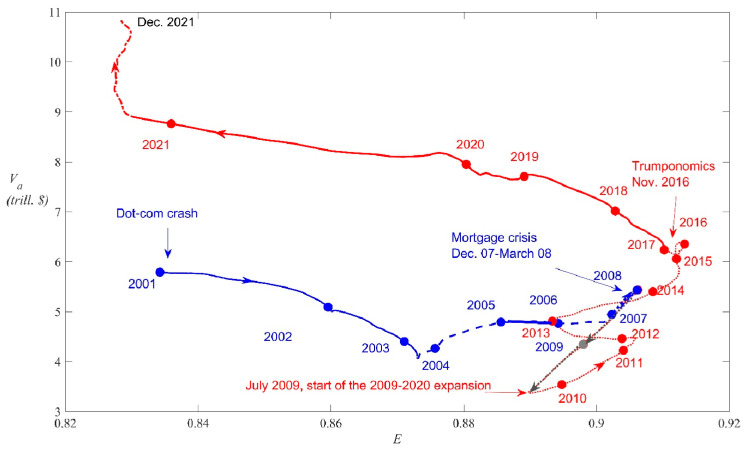
The trajectory of CPI adjusted total market value, *V_a_*, as a function of *E* from 2001 to 2021 [41]. The portion in blue corresponds to the period between the dot-com crash and the Mortgage crisis (2001–2008). The red portion corresponds to the 2009–2020 expansion. Full lines correspond to entire periods in which *V_a_* and *E* moved in opposite directions (see text). Dashed lines correspond to periods in which *V_a_* and *E* moved in the same direction. The dotted section corresponds to the erratic period whose start coincided with the Mortgage crisis. The dot-dashed section in the upper left corresponds to the last three quarters of 2021 in which the market entered in a phase of almost vertical growth of *V_a_*.

It is possible to identify some landmark events. For instance, the dot-com crash in 2001 seems to have triggered a process until 2003 in which *V_a_* steadily decreased and *E* increased quickly. Conversely, from 2017 to 2021 *V_a_* increased and *E* decreased fast and steadily. The start of this second period coincided with the advent of “Trumponomics”. The term refers to the economic policies of U.S. President Donald Trump, who won the 8 November 2016 presidential election on the back of bold economic promises to cut personal and corporate taxes, restructure trade deals and introduce large fiscal stimulus measures [47]. The period 2008–2015 (dotted curve), whose start coincided with the “Subprime Mortgage Crisis” or the “Mortgage crisis”, was quite erratic from the point of view of *V_a_* vs. *E.*

In summary, after 21 years, the market evenness roughly returned to the value it had in 2001, but the total market value doubled in CPI-adjusted dollars. This growth occurred entirely in the second half of the period, characterized by a process of concentration in which the evenness lost everything it had gained in the first half.

The negative relationship between the two productivity metrics with the market evenness agrees to what is often observed in ecological communities across different taxa. For example, the analysis of data from a large multi-site grassland experiment revealed that for plots which started with the same and even species composition, but which diverged in evenness over time, those with lower evenness attained a significantly greater biomass [48]. Moreover, the relationship between evenness and biomass across all plots in these experiments was also negative. The same was observed in other grassland experiments involving many plots of perennial grass species [49]. There are also examples of mixtures of species that converge with time towards a state of higher biomass and lower evenness for protozoa [50] and algae [51].

## 4. Explaining the Relationship between Productivity and Evenness in Stock Markets from a Community Ecology Perspective

Species interactions involve a complex balance of competition and facilitation in which indirect interactions occur if a third species (or more species) modifies the interaction between two other species [52]. It was argued that the success of species in a community is affected not only by direct interactions between species, but also by indirect interactions among groups of species [53,54]. The Lotka–Volterra generalized equations can naturally implement these indirect interactions through combinations of several pairwise interaction coefficients. That is, species *i* affects directly species *k* through the coefficient *α_ki_*, but also indirectly through the combination of *α_ji_* and *α_kj_* (i.e., species *i* affects directly species *j*, which in turn affects directly species *k*).

Regarding markets as ecological communities enables us to use the general machinery of the theory of community ecology [9] to understand the observed negative relationship between productivity, measured by *V_a_* and *R_a_*, and evenness, *E*. We will focus in particular to the Lotka–Volterra generalized equations and the interaction matrix, *α_ij_*, quantifying the strength of the effects between pairs of species.

To estimate the interaction matrix *α_ij_* of an ecological community a far from trivial task. A straightforward procedure is through pairwise competition trials by comparing the species yields in biculture relative to monoculture [50,55]. However, these experiments, which are common in community ecology and agricultural science are feasible for a small number of coexisting species *S* [17,37,49,50]. This is because the number of required experiments grows as *S*^2^. Furthermore, such experiments are not feasible in markets since one cannot isolate companies from the rest of the market to study their evolution under controlled conditions. Hence, we have to make use of theoretical analysis in terms of in silico synthetic communities.

### 4.1. Ensemble of Synthetic Communities

Therefore, let us use an approach based on Robert May’s theoretical work in community ecology in terms of randomly assembled communities [38]. The idea is to consider an ensemble of pairwise interaction matrices whose diagonal elements, corresponding to intraspecific interactions of each species *i*, are set to −1, as it is customarily performed in community ecology [38]. The off-diagonal matrix elements, corresponding to interspecific interactions between different species *i* and *j*, are drawn from a uniform random distribution centered around 0 and with radius *δ*, which can thus be interpreted as the intensity of interspecific interactions. That, is:(6)$$\alpha_{ij}=\left\{ \begin{array}{ll} -1 & \mathrm{if}\;i=j\\ \mathrm{random}\;\mathrm{in}\;[-\delta ,+\delta ] & \mathrm{if}\;i\neq j \end{array}\right.$$

Notice that the mean of the interspecific interaction coefficients is *μ* = 0, i.e., negative and positive interaction coefficients are equally likely. In fact, complex combinations of negative and positive interactions have been identified in a number of different ecological communities, like plant communities [49,56], freshwater communities [57], etc. The heterogeneity of interspecific interactions is controlled by *δ* since the variance of the interspecific interaction coefficients is given by *δ*^2^ = *δ*^2^/3, i.e., the greater *δ*, the greater the variance of interspecific interactions. To analyze the effect of varying the heterogeneity of interspecific interactions, we took into account that systems in which interspecific interactions are stronger than intraspecific interactions are likely to be unstable [58], thus we kept *δ* < 1. Thus, the parameter *δ* was varied from 0 to 0.9 in steps of 0.1.

To set the intrinsic growth rates, *r_i_*, it was used that, by Equation (1), on average *r_i_* is equal to the mean relative returns (*v_i_*(*t* + 1) − *v_i_*(*t*))/*v_i_*(*t*). Therefore, let us take *r_i_* = 0.014 for all *i* which is the mean of the empirically observed relative returns (This mean implies a double average, over companies and over time.). Next, to solve the Lotka–Volterra Equation (1) for each value of *δ*, 1000 simulations were run, each one starting from a random initial condition:(7)$$v_{i}(1)=\mathrm{random}\;\mathrm{in}\;[0,1],\;i=1,2,\ldots ,78.$$

The initial total market value, and the evenness are thus given, respectively, by: *V*(1) ≈ 78 × 0.5 = 39 (in arbitrary units) and *E*(1) ≈ 0.957 (see Appendix B). Notice that these initial values are close to the equilibrium values for *δ* = 0, *V*_0_* = 39 and *E*_0_* = 1(see Appendix C). From this initial arbitrary state, the transient dynamics towards equilibrium was studied. It is important to note that randomness only enters in the initial choice of the interspecific coefficients *α_ij_*, which then define a particular community by Equation (6), and in the initial configuration (i.e., Equation (7)). For each simulation the subsequent dynamics is strictly deterministic, and the community specified by a random interaction matrix given by Equation (6), in general does not allow for the coexistence of all the 78 species. Instead, some species extinguish with time; the coexistence of the 78 species is in general unfeasible for random matrices [37]. Therefore, simulations were stopped for a time, *T*, for which the first species extinguished (A species is considered extinguished when its biomass drops below a cutoff *v*_min_ << 1 (here I use *v*_min_ =10^−5^)). For small values of *δ*, *T* can be quite large (thousands of time steps). However, as *δ* increases, *T* decreases, until *T*~30 for *δ* = 0.9. Hence, to use the same simulation cutoff time for all values of *δ*, *T* = 30 was fixed (qualitatively similar results were obtained for smaller values of *T*, as shown in Appendix C).

The results of simulations are shown in Figure 5. For *δ* = 0.1, only slight deviations in *V* and *E* from their initial values, *V*(1) = 39 and *E*(1) = 0.957, occur (Figure 5a). As *δ* increases, the community moves towards higher values of *V* and lower values of *E*. Indeed, the curves $\bar{E}(\delta )\;\mathrm{and}\;\bar{V}[\delta ]$ (the bars denote average over simulations) appear to be mirror images of each other.

These opposite trends for $\bar{E}(\delta )\;\mathrm{and}\;\bar{V}[\delta ]$ can be understood as follows. It is immediate that the evenness will decline when *δ* is increased. This is because for *δ* = 0 the interaction matrix *α_ij_*, given by Equation (6), reduces to the identity matrix, and then all distinction among the companies disappears, Therefore, the evenness tends to its maximum possible value of *E*_0_* = 1 (see Appendix C). As the interactions between companies are “turned on” (*δ* > 0), the equivalence between companies breaks down and the system departs from this state of maximum evenness. The larger the heterogeneity (variance) of these interactions the larger this departure. A derivation that $\bar{V}$[*δ*] is a monotonic increasing function of *δ* requires a little bit more of algebra. In a nutshell the idea is that, even though by Equation (6) positive and negative interspecific interactions are equally likely, the effect of positive interactions outweighs the effect of negative interactions, as it is shown in Appendix C. Moreover, it can also be derived that the time derivative of $\bar{V}$[*δ*], i.e., $\bar{R}$[*δ*], is a monotonic increasing function of *δ*.

Therefore, a way to generate the observed inverse relationship between productivity metrics and *E*, is by changing the interaction strength between companies: if *δ* increases (decreases) *R* and *V* tend to increase (decrease) and simultaneously *E* tends to do the opposite, i.e., to decline (rise).

Regarding the mechanism promoting the growth of *δ*, and ultimately behind the negative relationship between productivity and evenness, monetary policy is a natural candidate. That is, when the monetary policy is loose and interest rates are low, capital flows to firms. This injection of money promotes the idea that firms address new business opportunities which multiply the interactions between them, either in the form of cooperation through new contracts, joint ventures, etc. or competition in new segments. Such an increase in the heterogeneity of interactions among companies is equivalent in our model to increase *δ*. Indeed, high productivity coincided with an expansion in money supply, M1 [59], and mainly with low effective interest rates (see Appendix D). The relationship between interest rates and evenness, or between money supply and evenness is less clear. Although at the beginning of the period the evenness soared with high interest rates, it persisted high during 2009–2015 when, in order to combat the Great Recession, the U.S. Federal Reserve ran a quantitative easing program and kept the effective interest rate at virtually zero [59] (see Appendix D). In a similar vein, it was observed that algal biovolume, a surrogate for biomass, increased, whereas evenness decreased with increasing total supply of resources in algal communities [60].

Two remarks are in order. Firstly, the monotonic curves $\bar{E}(\delta ),\bar{R}[\delta ]\;\mathrm{and}\;\bar{V}[\delta ]$ of Figure 5 were obtained as averages over 1000 simulations. Nevertheless, this does not imply that if *δ*_1_ < *δ*_2_ all simulations performed with *δ*_1_ will produce a *V* smaller than the one produced by all simulation with *δ*_2_ or an *E* larger than the one produced by all simulation with *δ*_2_. Hence, this approach is also able to yield periods in which *R* and/or *V* move in the same direction as *E* (either both upward or downward), but they will be less likely than periods in which productivity metrics and *E* move in opposite directions. This is in agreement with what is shown in Figure 4 for the empirical trajectory of *V* vs. *E*: those sections in which both variables move in the same direction are rarer and shorter (e.g., during 2004).

Secondly, this approach, in terms of random matrices, produces only qualitative evidence for the observed *V* vs. *E* trend in the U.S. stock market. To obtain a better quantitative description, one has to consider more complicated structured interaction matrices. This issue is beyond the scope of this study, but some recent advances are briefly reviewed in the next subsection. Indeed, the random matrices approach, which is commonly used in various fields such as physics, mathematics, and finance, has certain limitations and restrictions. A main restriction is its assumptions of randomness. The random matrices approach relies on the assumption that the matrix elements are independent and identically distributed random variables. However, in some real-world scenarios, this assumption may not hold true. Real-world data often exhibit correlations, dependencies, or non-random patterns that may not be accurately captured by random matrices.

### 4.2. Other Ecologically Based Approaches Supporting the Negative Relationship between Productivity and Evenness

The classical *Ecological Niche Theory* (ENT) states that an ecological community is made up of a limited number of niches, each occupied by a single species and that differences among species in their niches are important in determining the outcome of species interactions as might be revealed in their distributions and/or abundances in ecosystems [61]. Using ENT, the pattern of increasing biomass accompanied by decreasing evenness was firstly mathematically derived for the case of pure competition [62], which implies a restriction of the general interactions of LVGT only to mutually competitive interactions for resources. More recently, this result was extended to the more realistic case of generalized interactions. This was performed through the so-called Lotka–Volterra Niche Game Model (LVNGM) [63], resulting from the combination of ENT and Game Theory. Other recent works approaching financial markets as ecosystems have contributed to support the generality of the inverse relationship between productivity and evenness. Indeed, population dynamic models can be used in conjunction with time series of species abundances to infer the interaction coefficients between companies through indirect methods. One of such indirect methods is the so-called *Pairwise Maximum-Entropy* (PME) modeling [64]. PME modeling is a particular implementation of the of Maximum Entropy general approach proposed by Jaynes [65,66] which has been used in finance for different purposes, like ranking the performance of mutual funds [67], retrieving the risk neutral density of asset returns [68], investigating the effect of size differences on cost efficiency heterogeneity in U.S. commercial banks [69], etc. In the last two decades, PME models have been used to analyze ecological data associated with diverse problems, such as animal flocks [70], and community ecology [37,71,72,73]. In fact, PME modeling has been applied for a subset of the US companies I consider here in two recent studies, each one focused on a different subject, across different time lengths or training periods *T_tr_*. The first one addressed the issue of inferring adjacency matrices defining the network that describe the interactions between firms in a fashion similar to how theoretical ecology pictures the interaction of species in an ecosystem [74]. A main finding of a community analysis on the resultant networks was that the network modules derived from a PME matrix, *M_ij_*, coincide almost exactly with the industry groupings of the firms defined by the *Global Industry Classification Standard* (GICS) [75]. The second study tested the combination of this PME approach with evolutionary game theory for quantitative market forecasting by taking *α_ij_* = *M_ij_* [7]. It turns out that the resulting forecasting method does a decent job of predicting empirical shares of the companies along several choices of validation periods. Interestingly, these interaction matrices *α_ij_* obtained by the PME method in [7,74] exhibit properties which are similar to the ones of the synthetic communities defined by Equation (6), namely that (a) most of its off-diagonal element are in the interval (−1, +1) and (b) with a mean close to 0.

## 5. Conclusions

As we have seen, regarding markets as ecosystems can be traced back to the late 1950s [3]. Since then, different authors have contributed to building this analogy and used it to gain insight into market forces. However, there has been a lack of quantitative tools so far useful to the practitioners [76]. Indeed, the main novelties of this study are as follows:

Firstly, it raises the productivity vs. diversity issue, a fundamental question of community ecology, in the context of financial markets modeling. It is worth mentioning that a similar conclusion was drawn using a different diversity measure provided by the largest eigenvalue of the correlation matrix among stocks [77]. The productivity–diversity tradeoff is important because, as it happens in ecology, in economics, decision makers need to strike a balance in resource allocation by considering both productivity-enhancing investments and maintaining a diverse to mitigate risks and promote long-term sustainability and resilience.

Secondly, it uses the Shannon evenness of market values to quantify the market diversity as opposed to market concentration. Being a normalized metric, the Shannon evenness is particularly useful when working with samples of companies of large markets such as NYSE. Additionally, it allows a quantitative comparison of the evenness among different markets or among different industrial sectors of the same market. The use of the Shannon evenness was instrumental to detect an important pattern of NYSE market dynamics between 2001 and 2021, namely the fact that high productivity was achieved when the evenness dropped; conversely when the evenness soared (during the business cycle 2001–2008) productivity plunged. Interestingly, such negatively correlated regime parallels the relationship between total biomass and species evenness observed in several ecosystems across distinct taxa (plants, algae, protozoa, etc.). In the case of economics and finance, balancing productivity and diversity is crucial for sustainable economic growth. High productivity can boost overall output and efficiency, leading to economic expansion. Diversity, on the other hand, can contribute to resilience and adaptability, allowing economies to better withstand disruptions. Diversity also plays a vital role in fostering innovation and creativity. When a system encompasses diverse perspectives, knowledge, and skill sets, it is more likely that it promotes the generation of new ideas and approaches. Recognizing the potential adverse effects of losing diversity, decision makers can implement policies that promote a more diverse economic landscape. This can involve supporting industries with growth potential, fostering entrepreneurship, encouraging small and medium-sized enterprises, and providing incentives for diverse business models and market entrants. Such policies can help maintain a resilient economy, reduce concentration risks, and encourage innovation and competition.

Thirdly, as far as the author knows, May’s model [38] has not been previously used to analyze the relationship between evenness and productivity, neither in ecology nor in economics. Specifically, the model allows to explain how an inverse relationship between productivity and diversity can emerge when loosening or tightening the monetary policy. This has profound implications for decision makers, who need to carefully balance the short-term benefits of loosening monetary policy, such as increased liquidity and economic stimulus, with the potential long-term undesired effects on the economy. While monetary easing may provide immediate economic boosts, it can also discourage productivity improvements and hinder the development of a diverse and resilient economy. In that sense, the above finding serves to assess the trade-offs and evaluate the long-term consequences of monetary policy decisions.

Let us conclude with some research directions that seem worth investigating in future works. One important issue is the generality of negative correlation between market productivity and market evenness. For example, one may wonder whether this pattern is a particularity of the US stock market or if it is shared by other stock markets in different countries? Thus, analyzing financial markets from other countries is a natural next step. Another question is how the detected pattern is connected to long-term trends in demographics and the inter-industry reconfiguration of firms away from traditional manufacturing [78]. The business ecosystem perspective is also useful to develop quantitative methods to forecast future market values of firms [79], or to define fitnesses for firms and disentangle the effects of selection and the environment in the evolutionary dynamics of financial markets [80].

## Figures and Tables

**Figure 1 entropy-25-01029-f001:**
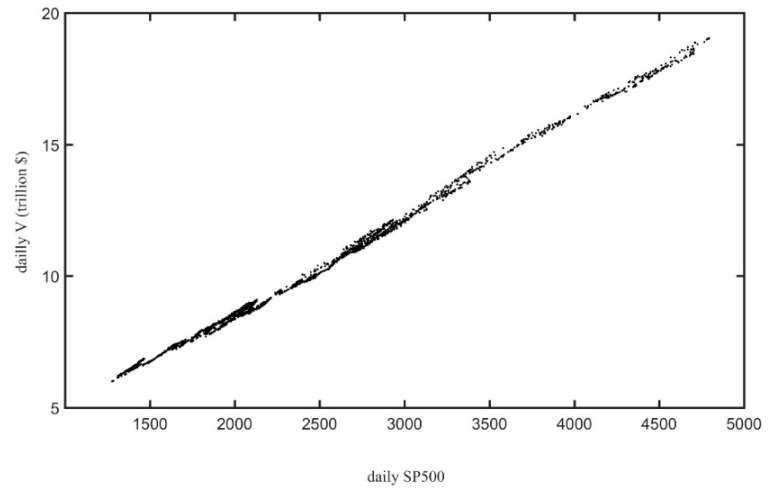
The total market value *V* vs. the S&P 500 index, for the period 2012–2021 [41].

**Figure 2 entropy-25-01029-f002:**
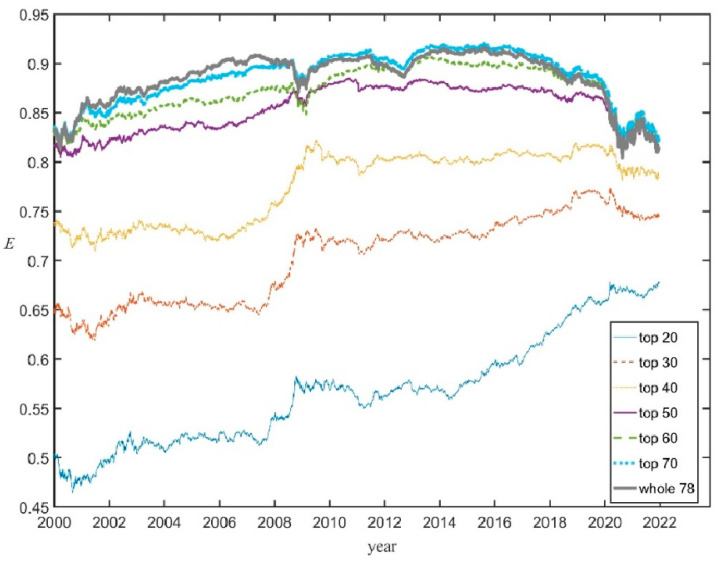
The Shannon evenness curves *E_N_* (*t*) computed for the top *N* = 20 companies, the top *N* = 30, …, the whole set of 78 companies (thick gray curve).

**Figure 5 entropy-25-01029-f005:**
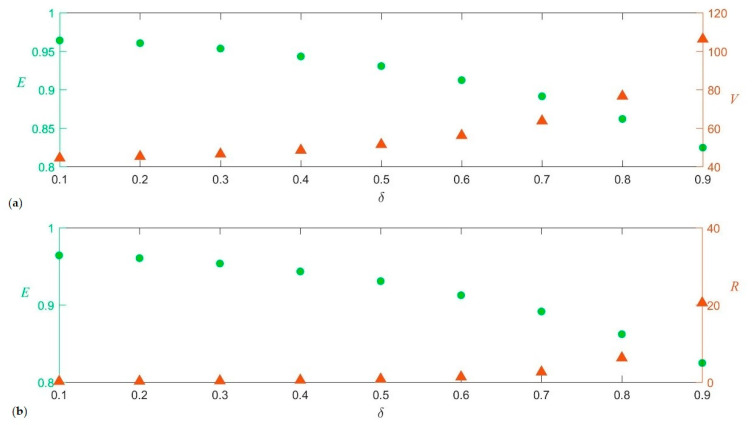
For each value of *δ* the corresponding *E*, *R* and *V* were obtained as averages $\bar{E},\bar{R}\;\mathrm{and}\;\bar{V}$ over 1000 simulations of synthetic communities, defined by the interaction matrix of Equation (6), starting with the initial random configuration of Equation (7). (**a**): Curves of $\bar{E}$ vs. *δ* (circles, left axis) and $\bar{V}$ vs. *δ* (triangles, right axis). (**b**): Curves of $\bar{E}$ vs. *δ* (circles, left axis) and $\bar{R}$ vs. *δ* (triangles, right axis).

**Table 1 entropy-25-01029-t001:** The 78 companies considered in this study ordered by their market value as of 31 December 2021 [41].

Company	Ticker	Market Val (USD Bill)	Rank	Sector	Industry
Apple	AAPL	2902	1	Technology	Consumer Electronics
Microsoft	MSFT	2522	2	Technology	Software–Infrastructure
Amazon	AMZN	1697	3	Consumer Cyclical	Internet Retail
Berkshire Hathaway	BRK	662.63	4	Financial Services	Insurance
JP Morgan	JPM	472.51	5	Financial Services	Banks
United Health Group	UNH	466.21	6	Healthcare	Healthcare Plans
Johnson & Johnson	JNJ	450.36	7	Healthcare	Drug Manufacturers
Home Depot	HD	433.37	8	Consumer Cyclical	Home Retail
Walmart	WMT	401.35	9	Consumer Defensive	Discount Stores
P&G	PG	392.11	10	Consumer Defensive	Household
Bank of America	BAC	359.38	11	Financial Services	Banks
Pfizer Inc.	PFE	331.86	12	Healthcare	Drug Manufacturers
The Walt Disney Company	DIS	281.54	13	Comm. Services	Entertainment
Cisco Systems, Inc.	CSCO	267.27	14	Technology	Comm. Equipment
Nike	NKE	263.55	15	Consumer Cyclical	Footwear and Access.
Thermo Fisher Scientific Inc.	TMO	263.18	16	Healthcare	Diagnosis and Research
Exxon Mobil	XOM	259.38	17	Energy	Oil and Gas
The Coca-Cola Company	KO	256.09	18	Consumer Defensive	Beverages
Costco	COST	251.74	19	Consumer Defensive	Discount Stores
Abbott Laboratories	ABT	248.28	20	Healthcare	Medical Devices
PepsiCo, Inc.	PEP	240.24	21	Consumer Defensive	Beverages
Oracle	ORCL	232.89	22	Technology	Software–Infrastructure
Comcast	CMCSA	228.16	23	Comm. Services	Telecom Services
Chevron	CVX	226.46	24	Energy	Oil and Gas
Verizon	VZ	218.12	25	Comm. Services	Telecom Services
Intel Corporation	INTC	209.6	26	Technology	Semiconductors
QUALCOMM Incorporated	QCOM	205.73	27	Technology	Semiconductors
Merck & Co., Inc.	MRK	193.72	28	Healthcare	Drug Manufacturers
Wells Fargo	WFC	186.44	29	Financial Services	Banks
Anthem	UPS	186.41	30	Industrials	Integrated Freight and Logistics
Lowe’s	LOW	174.15	31	Consumer Cyclical	Home Retail
Morgan Stanley	MS	173.96	32	Financial Services	Banks
Honeywell International Inc.	HON	142.79	33	Industrials	Conglomerates
CVS Caremark	CVS	136.38	34	Healthcare	Healthcare Plans
Bristol-Myers Squibb Company	BMY	134.24	35	Healthcare	Drug Manufacturers
AT&T	T	132.58	36	Comm. Services	Telecom Services
Raytheon Technologies Corp.	RTX	128.51	37	Industrials	Aerospace and Defense
The Goldman Sachs Group, Inc.	GS	127.61	38	Financial Services	Banks
American Express Company	AXP	124.5	39	Financial Services	Credit Services
IBM	IBM	120.04	40	Technology	Information Tech. Serv.
Citigroup	C	119.84	41	Financial Services	Banks
Boeing	BA	118.56	42	Industrials	Aerospace and Defense
Target	TGT	110.89	43	Consumer Defensive	Discount Stores
Caterpillar Inc.	CAT	110.79	44	Industrials	Farm and Heavy Constr.
Deere & Company	DE	105.68	45	Industrials	Farm and Heavy Constr.
General electrics	GE	103.83	46	Industrials	Specialty Industr. Machinery
3M Company	MMM	101.58	47	Industrials	Conglomerates
Lockheed Martin Corporation	LMT	96.32	48	Industrials	Aerospace and Defense
ConocoPhillips	COP	94	49	Energy	Oil and Gas
Phillips 66	TJX	90.56	50	Energy	Oil and Gas
Ford Motors	F	85.59	51	Consumer Cyclical	Auto Manufacturers
Cigna Corporation	CI	74.16	52	Healthcare	Healthcare Plans
FedEx Corporation	FDX	68.53	53	Industrials	Integrated Freight and Logistics
Northrop Grumman Corp.	NOC	60.49	54	Industrials	Aerospace and Defense
Capital One Financial Corp.	COF	60.05	55	Financial Services	Credit Services
The Progressive Corporation	PGR	59.99	56	Financial Services	Insurance
Humana Inc.	HUM	59.75	57	Healthcare	Healthcare Plans
General Dynamics	GD	57.88	58	Industrials	Aerospace and Defense
Enterprise Products Partners L.P.	EPD	47.79	59	Energy	Oil and Gas
AIG	AIG	46.55	60	Financial Services	Insurance
Walgreens Boots Alliance	WBA	45.03	61	Healthcare	Pharmaceutical Retailers
HP Inc.	HPQ	40.79	62	Technology	Computer Hardware
Exelon Corporation	EXC	40.34	63	Utilities	Utilities-Regulated Electric
Sysco Corporation	SYY	40.27	64	Consumer Defensive	Food Distribution
Archer-Daniels-Midland Comp.	ADM	37.85	65	Consumer Defensive	Farm Products
The Travelers Companies, Inc.	TRV	37.73	66	Financial Services	Insurance
McKesson Corp.	MCK	37.24	67	Healthcare	Medical Distribution
The Kroger Co.	KR	33.28	68	Consumer Defensive	Grocery Stores
The Allstate Corporation	ALL	33.06	69	Financial Services	Insurance
Tyson Foods, Inc.	TSN	31.65	70	Consumer Defensive	Farm Products
Nucor Corporation	NUE	31.1	71	Basic Materials	Steel
Valero Energy	VLO	30.73	72	Energy	Oil and Gas
AmerisourceBergen	ABC	27.78	73	Healthcare	Medical Distribution
Best Buy Co., Inc.	BBY	24.44	74	Consumer Cyclical	Specialty Retail
Cardinal Health	CAH	14.26	75	Healthcare	Medical Distribution
Arrow Electronics, Inc.	ARW	9.14	76	Technology	Electronics Distribution
Fannie Mae	FNMA	0.95	77	Financial Services	Mortgage Finance
Chico’s FAS, Inc.	CHS	0.66	78	Consumer Cyclical	Apparel Retail

## Data Availability

The data that support the findings of this study are available from the corresponding author upon request.

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
