# Peer review of "Productivity vs. Evenness in the U.S. Financial Market: A Business Ecosystem Perspective"

_entropy, 2023, doi:10.3390/e25071029_

Round 1

Reviewer 1 Report

Once again, I must thank the Editorial Board of "Entropy" for giving me the opportunity to serve as a reviewer for their famed publication. Now focused on the review of the article entitled: "Productivity vs. evenness in the U.S. financial market: A business ecosystem perspective" (id: entropy-2459424), the following is the result of my review:

Major revision

Indeed, it is not a bad paper, but there are some limitations that make not advisable to accept the paper in its current state. Below I indicate a series of proposals for improvement, changes and suggestions, which the authors should follow:

- The abstract is within the word limit established for this publication. However, please eliminate any type of formula or variable in the elaboration of the abstract. Please express succinctly the content of the paper but avoid such specification.

- All tables and figures must contain the original source.

- I am afraid that the data source may be somewhat out of date:

- Fortune Magazine. https://fortune.com/rankings/ Accessed November 09, 2022

Two options: could you obtain more recent data from this source and replicate the study? If not, please mention this limitation in the conclusion, as well as any other you may have encountered during the course of this work.

Regarding the antiquity of the data, Table 1 "The 78 companies considered in this study ordered by their market value as of December 31, 2021"...

In any case, in this table also determine the sector to which each listed company belongs and insert some elementary descriptive statistics.

Finally, I would like to point out that the article is quite well structured and that the level of academic English used is quite acceptable. However, part of the paper is written in "first-person". I provide, I used, I resort, ... I would like the authors to understand that Entropy is an academic journal, not a personal diary, hence I suggest that they use non-personal forms in the redrafting of the article.

With my best wishes in your personal and academic life,

The reviewer

Good 

Author Response

Response to Reviewer 1 Comments

Indeed, it is not a bad paper, but there are some limitations that make not advisable to accept the paper in its current state. Below I indicate a series of proposals for improvement, changes and suggestions, which the authors should follow:

- The abstract is within the word limit established for this publication. However, please eliminate any type of formula or variable in the elaboration of the abstract. Please express succinctly the content of the paper but avoid such specification.

Response: Done

- All tables and figures must contain the original source.

R: Done.

- I am afraid that the data source may be somewhat out of date:

 Fortune Magazine. https://fortune.com/rankings/ Accessed November 09, 2022

R: The Fortune 100 list changes every year. The above link contains the list I used.

Two options: could you obtain more recent data from this source and replicate the study? If not, please mention this limitation in the conclusion, as well as any other you may have encountered during the course of this work.

Regarding the antiquity of the data, Table 1 "The 78 companies considered in this study ordered by their market value as of December 31, 2021"...

R: I bought the data in September 2022, so they covered from January 1st 2000 to September 2022. I discarded data corresponding to the incomplete year 2022 and kept the 22 years = 5536 market days from January 1st 2000 to December 31 2021. This is a quite long period. In particular it covers three recessions, i.e. and thus two business cycles. I mention this in the paper. At any event, I have checked that the discarded data do not affect the results.

In any case, in this table also determine the sector to which each listed company belongs and insert some elementary descriptive statistics.

R: Thank you for this good advice. I added two columns to Table 1 with, respectively, the Sector and the Industry to which the company belongs to.

Finally, I would like to point out that the article is quite well structured and that the level of academic English used is quite acceptable. However, part of the paper is written in "first-person". I provide, I used, I resort, ... I would like the authors to understand that Entropy is an academic journal, not a personal diary, hence I suggest that they use non-personal forms in the redrafting of the article.

R: Done

Reviewer 2 Report

The material presented in this manuscript is quite interesting and it suits well to Entropy. Some revisions are however necessary:
- something goes wrong with the summation index in Eq.(3)
- the periods selected for evaluating the mean of R in Fig.3 look somewhat arbitrary and seem to be adjusted to fit the predetermined thesis
- from 'Dynamics of competition between collectivity and noise in the stock market' Physica A 287 (2000) 440 a similar conclusion is drawn. There a different (but largely equivalent) measure of evenness is used (the largest eigenvalue of the correlation matrix) and the result is that the market decreases are more collective than increases.  
- can the simplicity of uniform random distribution used as given by Eq.(6) (as in May's [38] theoretical/model work) be justified for the present stock market considerations? From Fig.80 in 'Physical approach to complex systems' Physics Reports 515 (2012) 115–226 a hierarchical organization of correlations in the same market can be anticipated.

A systematic proofreading is needed as there are many misprints/omissions.

Author Response

Response to Reviewer 2 Comments

The material presented in this manuscript is quite interesting and it suits well to Entropy. Some revisions are however necessary:

- something goes wrong with the summation index in Eq.(3)

Response: Thank you for pointing this error. I deleted the company index 'i' to the left, total returns R is a global quantity, i.e. defined for the whole set of companies.

- the periods selected for evaluating the mean of R in Fig.3 look somewhat arbitrary and seem to be adjusted to fit the predetermined thesis

R: In fact, as mentioned, the three portions were selected according to the behavior displayed by the evenness E, that is: growing E, oscillations of E around a high value, and decreasing E. Please look at the green line, corresponding to the E curve in Fig.3, and see lines 254-262.

- from 'Dynamics of competition between collectivity and noise in the stock market' Physica A 287 (2000) 440 a similar conclusion is drawn. There a different (but largely equivalent) measure of evenness is used (the largest eigenvalue of the correlation matrix) and the result is that the market decreases are more collective than increases.

R: Thank you for pointing this work. I mention it in the Conclusions of the revised manuscript the agreement with these results. 

- can the simplicity of uniform random distribution used as given by Eq.(6) (as in May's [38] theoretical/model work) be justified for the present stock market considerations? From Fig.80 in 'Physical approach to complex systems' Physics Reports 515 (2012) 115–226 a hierarchical organization of correlations in the same market can be anticipated.

R: This is an interesting observation. May’s model has been highly influential in theoretical ecology.

The main virtue of May's model is that, despite its simplicity and generality, its ability to capture fundamental dynamics, generate quantitative predictions, and offer insights into ecological communities' structure. The model has also been applied to other fields beyond theoretical ecology, including statistical physics, network theory, and complex systems, to study phenomena characterized by complex interactions and dependencies. The model's versatility and potential for cross-disciplinary applications contribute to its virtues and continued relevance. Nevertheless, as you mention, it is important to note that while the random matrix model has made significant contributions to theoretical ecology, it is a simplification of real-world ecological systems. The May’s model assumes idealized conditions, which may limit its applicability to complex and specific ecological contexts. I warn about this in the revised version of the manuscript (see P. 473-480).

Reviewer 3 Report

1. The Abstract should be longer. It should contain the short explanation, materials, source of data, methods, results and conclusions.

2. Where is the explanation why this subject has been chosen?

3. I will suggest to explain what does a business ecosystem perspective is.

4. What are the criteria for selecting this methodology?

5. What are the restrictions of the used methods?

6. Please, indicate the details: 

- "along the six years", which years? (line169),

- after 21 years the market .... (line 245),

- accross different taxa - what is it (line 250),

7. Please transform conclusions directly resulting from the analyses of statistical methods into conclusions of an applied nature.

8.  What are the first and second order conclusions of the analyses carried out.

9. Where are the hypotheses?

10. Where, in the conclusion, it is proven that hypotheses have been confirmed or rejected.

11.  What are the authors’ suggestions to reduce the limitations of the study in order to continue it.

Dear Author,

 in my opinion the quality of English is fine. 

Best 

the Reviewer

Author Response

Response to Reviewer 3 Comments

  1. The Abstract should be longer. It should contain the short explanation, materials, source of data, methods, results and conclusions.

Response: Please notice that Entropy limits the length of the abstract to a maximum of 200 words.

In addition, the Reviewer 1 suggested to express succinctly the content of the paper in the Abstract but avoiding such specification. the opposite, i.e to reduce the length of the abstract.

As a compromise solution I kept the length but included most of the points you mentioned above.

  1. Where is the explanation why this subject has been chosen?

R: It seems I failed to explain this with clarity. In a nutshell the idea is, using the analogy between markets and ecosystems, allows to use several powerful techniques from theoretical ecology into economics and finance to get insights into the market dynamics. In this work I focus on explaining the inverse relationship I detected between productivity and evenness in a set including the largest companies in the U.S. stock market, using both empirical evidences and theoretical modeling. This relationship has crucial implications in the function of both, markets and ecosystems. In the revised version I added a paragraph explaining the point (lines 77-85).

  1. I will suggest to explain what does a business ecosystem perspective is.

R: Thank you for the suggestion. In this revised version I explain the business ecosystem perspective in more detail. Please see the lines 106-126.

  1. What are the criteria for selecting this methodology?

R: The approach I follow in this work started from an empirical observation of a negative relationship between evenness and productivity in the NYSE market. It turns out that the same relationship has been found in several ecological communities, involving algae, plants, etc. So the next step was, using the analogy between markets and ecosystems, try to understand the possible origin of this negative relationship. Therefore, I used an approach introduced in theoretical ecology by Robert May in 1972 to explore the dynamics of ecosystems, which consists in representing the interspecific interactions as random matrices. And hence I show that it provides an explanation in terms of the strength of these pairwise interactions. Please see lines 359-364 and 473-474.

  1. What are the restrictions of the used methods?

R: I mention in the Conclusions that an important issue is about the generality of negative correlation between market productivity and market evenness. For example, one may wonder whether this pattern is a particularity of the US stock market or if it is shared by other stock markets in different countries? Thus, analyzing financial markets from other countries is a natural next step. Another question is how the detected pattern is connected to long-term trends in demographics and the inter-industry reconfiguration of firms away from traditional manufacturing as pointed out by Triplett et al. 2022 (ref. [78]).

In addition, the random matrices approach, which is commonly used in various fields such as physics, mathematics, and finance, has certain limitations and restrictions. A main restriction is its assumptions of randomness. The random matrices approach relies on the assumption that the matrix elements are independent and identically distributed random variables. However, in some real-world scenarios, this assumption may not hold true. Real-world data often exhibit correlations, dependencies, or non-random patterns that may not be accurately captured by the random matrices framework. I warn about this in the revised version of the manuscript (see lines 473-480).

  1. Please, indicate the details:

- "along the six years", which years? (line169),

R: Sorry, my mistake. The period is 21 years, from 2001 to 2021.

- after 21 years the market .... (line 245),

R: Please see the above response.

- accross different taxa - what is it (line 250),

R: Taxa is the plural of taxon, which denotes any unit used in biological classification, or taxonomy. Here I mean for example: algae, plants, etc.

  1. Please transform conclusions directly resulting from the analyses of statistical methods into conclusions of an applied nature.

R: This is also good advise. In the revised version I added paragraphs explaining the practical implications of each conclusion and how they can help decision-makers. Please, see lines 533-537, 550-564, and 568-578.

  1. What are the first and second order conclusions of the analyses carried out.

 R: Please see the above response.

  1. Where are the hypotheses?

R: Actually, in this article I follow a kind of inductive approach rather than a deductive one. In the deductive method, a hypothesis is formulated as a starting point or premise from which logical deductions are made. The hypothesis is a statement or proposition that serves as the foundation for the deductive reasoning process. The deductive method seeks to test the validity of the hypothesis by applying logical rules of inference.

On the other hand, the inductive approach starts with specific observations or data (in this study, the negative relationship between evenness and productivity) and looks for patterns, trends, or regularities based on the observations with the coal of formulating a general conclusion or explanation based on the patterns observed. As I understand inductive reasoning, it aims to provide probable or likely conclusions that are supported by the available evidence (e.g. the strength of the interactions between companies which can be interpreted as reflecting the looseness of monetary policy).
I apologize for being so long in my explanation, but you raised a very interesting point.

  1. Where, in the conclusion, it is proven that hypotheses have been confirmed or rejected.

R: Please see the above response.

  1. What are the authors’ suggestions to reduce the limitations of the study in order to continue it.

R: I think including this is good advice, it will help to develop the approach to markets as biological communities, i.e. the business ecosystem perspective (please see responses above on this specific point). For example, in a different paper we have just submitted we use the business ecosystem perspective as a quantitative method to forecast future market values of firms, or to define fitnesses for firms and disentangle the effects of selection and the environment in the evolutionary dynamics of financial markets. I mention this in the revised version of the manuscript (see lines 588-591). I also added a couple of new references ([79] and [80]).

Round 2

Reviewer 1 Report

For my part, I consider that the article could be accepted (regardless of the fact that the answer given by the author about the antiquity of the data, in my opinion, is not particularly convincing).

Therefore, it only remains for me to congratulate the author.

With my best wishes in your personal and academic life,

The reviewer

Reviewer 2 Report

The revisions are satisfactory and the manuscript thus ready for publication in its current form.

Reviewer 3 Report

Dear Author,

I am happy to read the corrected version of Your manuscript.

It is always a hard work when we know that the whole process of preparing it relays on single person.

In my opinion the manuscript can be published now.

Best reagards

Your Reviewer

Dear Author,

 in my opinion, the quality of English is all right.